# Physicochemical Aspects, Bioactive Compounds, Phenolic Profile and In Vitro Antioxidant Activity of Tropical Red Fruits and Their Blend

**DOI:** 10.3390/molecules28124866

**Published:** 2023-06-20

**Authors:** Yaroslávia Ferreira Paiva, Rossana Maria Feitosa de Figueirêdo, Alexandre José de Melo Queiroz, Lumara Tatiely Santos Amadeu, Francislaine Suelia dos Santos, Carolaine Gomes dos Reis, Ana Júlia de Brito Araújo Carvalho, Marcos dos Santos Lima, Antônio Gilson Barbosa de Lima, Josivanda Palmeira Gomes, Rodrigo Leite Moura, Henrique Valentim Moura, Eugênia Telis de Vilela Silva

**Affiliations:** 1Science and Technology Center, Federal University of Campina Grande, Campina Grande 58429-900, Brazil; antonio.gilson@ufcg.edu.br (A.G.B.d.L.); mourarodrigoleite@gmail.com (R.L.M.); 2Department of Agricultural Engineering, Federal University of Campina Grande, Campina Grande 58429-900, Brazil; rossanamff@gmail.com (R.M.F.d.F.); alexandrejmq@gmail.com (A.J.d.M.Q.); lumaratatielyea@gmail.com (L.T.S.A.); francislainesuelis@gmail.com (F.S.d.S.); carolainetecalimentos@gmail.com (C.G.d.R.); josivanda@gmail.com (J.P.G.); valentim_henrique@hotmail.com (H.V.M.); eugenia_telys@hotmail.com (E.T.d.V.S.); 3Department of Food Technology, Federal Institute of Sertão Pernambucano, Petrolina 56314-522, Brazil; ana.julia@ifsertao-pe.edu.br (A.J.d.B.A.C.); marcos.santos@ifsertao-pe.edu.br (M.d.S.L.)

**Keywords:** *Malpighia emarginata*, *Psidium guajava*, *Eugenia uniflora*, flavonoids, carotenoids, Pearson correlation

## Abstract

The combination of fruit pulps from different species, in addition to multiplying the offer of flavors, aromas and textures, favors the nutritional spectrum and the diversity of bioactive principles. The objective was to evaluate and compare the physicochemical characteristics, bioactive compounds, profile of phenolic compounds and in vitro antioxidant activity of pulps of three species of tropical red fruits (acerola, guava and pitanga) and of the blend produced from the combination. The pulps showed significant values of bioactive compounds, with emphasis on acerola, which had the highest levels in all parameters, except for lycopene, with the highest content in pitanga pulp. Nineteen phenolic compounds were identified, being phenolic acids, flavanols, anthocyanin and stilbene; of these, eighteen were quantified in acerola, nine in guava, twelve in pitanga and fourteen in the blend. The blend combined positive characteristics conferred by the individual pulps, with low pH favorable for conservation, high levels of total soluble solids and sugars, greater diversity of phenolic compounds and antioxidant activity close to that of acerola pulp. Pearson’s correlation between antioxidant activity and ascorbic acid content, total phenolic compounds, flavonoids, anthocyanins and carotenoids for the samples were positive, indicating their use as a source of bioactive compounds.

## 1. Introduction

Tropical fruits are some of the most appreciated natural foods, at first for their sensory attributes, but also for their nutritional contribution, mainly in terms of vitamins, minerals and bioactive principles. The variety of flavors and aromas provided by in natura presentations can be further multiplied through appropriate handling and processing, offering the consumer new sensory experiences and greater nutritional diversity.

The demand for practicality with good flavors, aromas and textures has increasingly replaced the consumption of fresh fruits with their derivatives [1]. In this segment, a practical and low-cost resource to offer new products is the production of blends of different types of fruits [2], resulting in products with peculiar palatability and acceptability criteria.

Brazil is the largest producer and exporter of acerola (*Malpighia emarginata*) in the world. This fruit stands out for its high content of ascorbic acid, as well as amino acids, minerals and bioactive compounds, such as carotenoids and phenolic compounds, which give it high antioxidant activity [3].

Brazil is also the world’s largest producer of guava (*Psidium guajava* L.), producing 424 thousand tons per year [4]. This fruit has large amounts of phytochemicals with antioxidant activity, as well as high levels of several vitamins and minerals [5]. Its high acceptance is due to its excellent sensory characteristics, mainly due to volatile compounds related to aroma [6].

Pitanga (*Eugenia uniflora* L.) is a native Brazilian species, rich in calcium, phosphorus, flavonoids, carotenoids and vitamin C, giving it high antioxidant activity. It has a predominantly acidic flavor, with variations observed in fruits with low acidity and specimens with high acidity, in addition to some astringency with a characteristic aroma [7].

Popular among consumers and indicated by health professionals, foods with properties considered functional have been used with the promise of preventing chronic diseases [8]. The combination of fruits of different species, complementing each other in the content of bioactive compounds, antioxidants and dietary fibers, presents itself as a food alternative with great functional potential, in addition to allowing the modulation of sensory attributes, offering unprecedented flavors and aromas.

Therefore, the objective of this study was to evaluate and compare the physicochemical characteristics, bioactive compounds, profile of phenolic compounds and in vitro antioxidant activity of pulps of three species of tropical red fruits (acerola, guava and pitanga) and of the blend produced from the combination of these pulps.

## 2. Results and Discussion

### 2.1. Physicochemical Characterization

Table 1 presents the results obtained for the physicochemical parameters of the tropical fruit pulps and the elaborated blend. The water contents of the pulps and blend ranged from 85.50 to 92.39%, with higher percentages in acerola and pitanga, not statistically different (*p* > 0.05) between them.

Reis et al. [9], when evaluating the water content of acerola pulps, identified an average value of 90.75 g/100 g, close to that identified in this work. Studying this same parameter in guava pulp, Maia et al. [10] found a value of 85.60 g/100 g, while Lins et al. [11] and Rivas et al. [12] reported mean values of 83.5 and 89.78 g/100 g, respectively, also close to those shown in Table 1. Reis et al. [13] found values of 90.06, 83.44 and 88.18 g/100 g in yellow, purple, and orange passion fruit pulp, respectively, levels close to those found in this work. Blends of pitaya with acerola combined in different proportions presented water contents ranging from 87.96 to 89.86 g/100 g [14] in the same range of value obtained in this study.

All pulps showed high water activity (a_w_ > 0.98), indicating a favorable substrate for the growth of microorganisms, biochemical reactions and, consequently, rapid deterioration. Close results were identified by Lemos et al. [15] both for acerola pulp (0.988) and for acerola and jabuticaba blends, which ranged from 0.988 to 0.993, and by Moraes et al. [14] in acerola and pitaya blends, ranging from 0.989 to 0.992.

Pitanga had the highest total titratable acidity (TTA) (3.42 g citric acid/100 g) and guava had the lowest (1.29 g citric acid/100 g), with the blend remaining close to the value determined in acerola (2.57 g citric acid /100 g). The pH determined in the acerola pulp is similar to that reported by Reis et al. [9] in the acerola variety ‘Flor Branca’, whose average value corresponded to 3.47, and by Magalhães et al. [16] when evaluating fruits of 20 accessions of acerola trees, in which they found an average value of 3.72, with variation in total acidity between 0.72 and 1.17 g/100 g. Both parameters meet the values recommended in Brazilian legislation, which indicates a minimum of 2.8 for pH and 0.8 g/100 g for total titratable acidity in acerola pulps [17]. For the other pulps, there is no prescription. Oliveira et al. [18], when evaluating Golden pitaya pulp, determined a pH of 4.79. Products with low pH have reduced enzymatic activity and microbial growth, meeting the preference of agroindustries [14].

TTA levels lower than those of the evaluated samples were quantified by Dantas et al. [19]. They evaluated the quality of pitangas during maturation, found acidity values ranging from 3.1 g/100 g in the golden-yellow stage to 2.3 g/100 g in the fully ripe fruit. Rivas et al. [12] found a pH of 3.70 and an acidity of 0.62 g/100 g in guava pulp. Schiassi et al. [20], when studying fruit pulps from the Brazilian Cerrado, determined acidity of 2.05 g/100 g in araçá (*Psidium guineense* Swartz), 0.47 g/100 g in buriti (*Mauritia flexuosa* L.), 0.64 g/100 g in cangaita (*Eugenia dysenterica* DC.), 1.66 g/100 g in yellow cajá (*Spondias mombin* L.), 0.90 g/100 g in mangaba (*Hancornia speciosa* Gomes) and 0.51 g/100 g in marolo (*Annona crassiflora* Mart.).

In the two formulations of mixed mango nectar (*Mangifera indica* L.) var. Carlota and passion fruit (*Passiflora setacea*) elaborated by Souza et al. [21], pH ranges of 3.60 and 3.70 were determined, which were close to that of the red fruit blend. 

Variations in acidity and fruit composition are explained by the influence of different cultivation methods, soil type, water and maturation stage, among others.

The samples differed statistically from each other (*p* < 0.05) in terms of ash, with the blend having a value 23.33% higher than pitanga, statistically equaling that of acerola.

Low ash values are common in fruits and derivatives, as observed in cashew pulp, pineapple, acerola, tangerine, pineapple with mint, guava, soursop, mango, grape, tamarind, passion fruit, cashew and plum, with values ranging from 0, 12 and 0.60 g/100 g [22]; in tamarillo pulp (*Solanum betaceum*) of the purple and yellow varieties, with 0.95 and 0.96 g/100 g, respectively [23]; and in frozen pineapple pulp, with 0.39 g/100 g [24]. Levels lower than those of the present study were determined in frozen acerola pulps produced and sold in Santarém (PA), which presented levels between 0.20 and 0.25 g/100 g [25]. The guava in the present study had a lower content than the guava pulp (1.31 g/100 g) studied by Srivastava et al. [26].

The total soluble solids (TSS) content of pitanga pulp differed statistically from the other samples (*p* < 0.05) and presented the lowest value, while the blend presented a content close to acerola and guava, with no significant difference (*p* > 0.05) between them. Ribeiro et al. [27], when analyzing an acerola blend with seriguela, reported a value of 11.43 °Brix, which is higher than that of the tropical fruit blend.

Acerola and guava pulps presented values higher than the minimum established by legislation for soluble solids, which are 5.50 and 7.0 °Brix, respectively [17]. Mariano-Nasser et al. [28], when evaluating the physicochemical characteristics of fruits of eight acerola cultivars, reported soluble solids values of 6.66 to 8.36 °Brix. Nazareno, Azevedo and Cardoso [29], assessing the quality of frozen fruit pulps from southwest Piauí, found average levels of 5.20 °Brix for acerola pulp and 4.80 °Brix for guava pulp. Pirola et al. [30] studied ripe pitanga fruits from the collection of native fruits at the Federal Technological University of Paraná for three years, obtaining average soluble solids of 9.39 °Brix in 2012, 7.12 °Brix in 2013 and 9.24 °Brix in 2015. It was evident that the physicochemical characteristics of fruit pulps vary according to the edaphoclimatic conditions of cultivation.

All evaluated samples differed statistically from each other (*p* < 0.05) in reducing sugar (RS) and total (TS) levels, with guava showing higher levels of both, increasing blend levels in relation to other pulps. 

For non-reducing sugars (NRS), guava reached the highest content again, while in pitanga, the value is not very expressive. Ribeiro et al. [27], evaluating acerola pulp and acerola blended with seriguela, found in acerola RS levels of 6.00 g glucose/100 g, TS of 6.87 g glucose/100 g and NRS of 0.87 g sucrose/100 g, and in the prepared blend, RS of 6.3 g glucose/100 g, AT of 8.3 g glucose/100 g and NRS of 2.0 g sucrose/100 g. Matos et al. [31], analyzing the same parameters, obtained results of 0.01 g glucose/100 g (RS), 3.26 g glucose/100 g (TS) and 3.08 g sucrose/100 g (NRS) for acerola pulp and 0.02 g glucose/100 g (RS), 5.30 g glucose/100 g (TS) and 5.02 g sucrose/100 g (NRS) for mixed pulp of acerola and jambolan.

In the guava juice evaluated by Freitas and Silva [32], the results were 4.51 g glucose/100 g for RS and 9.04 g sucrose/100 g for NRS; in guava nectar, the levels presented were 5.82 g glucose/100 g for RS and 10.40 g sucrose/100 g for NRS. In the mixed drink of guava and coconut water, studied by Shigematsu et al. [33], RS ranged from 2.75 to 4.12 g glucose/100 g, TS ranged from 4.12 to 7.59 g glucose/100 g and NRS ranged from 1.46 to 3.54 g sucrose/100 g.

So, given these levels of sugar in tropical fruit pulps and their blend when compared with the cited data from the literature, there were large amplitudes in the levels demonstrating great variability, probably due to the region of cultivation, variety, maturation stage and mix of fruits, among other factors.

### 2.2. Bioactive Compounds

The levels of bioactive compounds in the pulps and in the blend are shown in Table 2. The pulps showed a significant presence of bioactive compounds, especially the acerola pulp, which showed the best results for most parameters, except lycopene. High values of total phenolic compounds and ascorbic acid are verified in the samples and among the pigments. Despite the apparently red color of the pulps, carotenoids stand out followed by anthocyanins, with lycopene showing low levels.

As for ascorbic acid (AA), all evaluated samples differed statistically from each other (*p* < 0.05) and had high levels indicating that they were rich in AA. The composition with acerola favored the blend, which reached a higher AA content than most fruits, apart from acerola itself. 

The amount of ascorbic acid in pitanga pulp is not specified in legislation, but a minimum of 800 mg/100 g is required for acerola pulp and 24 mg/100 g for guava pulp [17]. These values were surpassed in the evaluated samples.

Lower AA values than those of the evaluated pulps were determined by some authors such as Malegori et al. [34] in mature acerola with a value of 2844 mg/100 g; by Nascimento et al. [35] in artisanal and industrialized acerola pulps that found values of 633.04 mg/100 g and 1080.11 mg/100 g, respectively; by Nazareno, Azevedo and Cardoso [29], evaluating pulp of acerola and guava, who reported values of 1015.42 mg/100 g and 29 mg/100 g, respectively. Poletto et al. [36] determined ascorbic acid levels in extracts of acerola by-products, identified as non-pomace (from the juice clarification step (centrifugation)), bagasse (seeds and peels) and lyophilized juice, which ranged from 158.2 to 193,7 mg/100 g. Helt, Navas and Gonçalves [37], evaluating pitangas produced in the region of Capão Bonito (SP), reported AA contents of 10.4 mg/100 g for orange pitanga and 8.9 mg/100 g for red pitanga. Evaluating the shelf life of a mixed drink made from coconut water and guava pulp, Shigematsu et al. [33] found AA levels ranging from 3 to 4.53 mg/100 g.

In the content of total phenolic compounds (TPC), the acerola pulp stood out from the others with a high content greater than 500 mg/100 g, and the guava and pitanga pulps presented medium contents of TPC between 100 and 500 mg EAG/100 g [38], resulting in a blend also with a high TPC content. Higher TPC contents than pitanga pulp were quantified by Silva et al. [39] for pitanga in natura (199.7 mg EAG/100 g) and pitanga pulp (219.7 mg EAG/100 g) and by Helt et al. [37], who verified levels of 435.3 mg EAG/100 g in orange pitanga and 1318.0 mg EAG/100 g in red pitanga. Variations in the TPC content for acerola pulp were also observed by Stafussa et al. [40], who presented a content of 13890.90 mg EAG/100 g, and by Fernandes et al. [41], who observed a content of 735 mg EAG/100 g in acerola juice. In guava, Omayio et al. [42], analyzing varieties from Kenya, reported values of 1649.14 EAG/100 g for the red-fleshed variety, 1386.54 EAG/100 g for the white-fleshed variety, and 1410.27 EAG/100 g for the strawberry variety. Foods rich in polyphenols are of interest to consumers due to their ability to prevent many chronic diseases [43].

In total flavonoids (TF), all samples differed statistically from each other (*p* < 0.05), with acerola showing the highest content (12.64 mg/100 g), followed by the blend (8.22 mg/100 g), favored by the high content provided by acerola, presenting twice the value determined in pitanga (4.05 mg/100 g) and approximately four times that of guava (1.91 mg/100 g). Higher levels of TF were quantified by Silva et al. [39] in samples of pitanga in natura with a value of 28.7 mg/100 g, in frozen pitanga pulp with a value of 28.0 mg/100 g, and in pitanga jelly with a value of 35.7 mg/100 g; by Omayio et al. [42], who reported values ranging from 188.25 to 250.66 mg/100 g for three different varieties of guavas produced in Kenya; and by Matos et al. [31], who determined a value of 16.16 mg/100 g in acerola pulp. Flavonoids are antioxidants and have antimutagenic, anti-inflammatory, antidiabetic, antimicrobial and antiviral effects [44].

Acerola had the highest anthocyanin contents, differing statistically from the others (*p* < 0.05), except for the blend, which again proved to be superior to pitanga and guava. Dantas et al. [19], evaluating the quality of pitanga fruits during maturation, found a value of 7.0 mg/100 g of anthocyanins in the ripe fruit, and Pereira et al. [45], in red pitanga pulp, reported a content of 2.14 mg/100 g, both higher than that quantified in our study.

Higher levels of anthocyanins were also verified by Zaicovski et al. [46] when performing the quantification of bioactive compounds in guavas bagged with fabrics of different colors, observing an anthocyanin content of 9.49 mg/100 g in the control sample (not bagged) and values ranging from 10.28 to 16.04 mg/100 g in the ones covered, concluding that the color of the packaging material influences the pigment content. Higher levels of anthocyanins were also verified by Lemos et al. [15], who, characterizing pulps, blends and jellies, observed values of 6.79 mg/100 g in acerola pulp and variations from 4.50 to 6.11 mg/100 g in acerola and jabuticaba blends in different proportions, including 1.52 mg/100 g in acerola jelly and from 1.56 to 1.96 mg/100 g in jellies of the blends.

The carotenoid contents of the three pulps and the blend differed statistically from each other (*p* < 0.05), with the highest value being determined in acerola, followed by the blend, which conferred higher values than those of pitanga and guava. All values were higher than those observed by Leiton-Ramírez et al. [47] in guava slices (1.34 mg/100 g bu), by Menezes et al. [48] in guava pulp (0.6082 mg/100 g) and by Viana et al. [49] in two hybrids and a commercial variety of acerola, which showed values of 0.016, 0.004 and 0.007 mg/100 g, respectively. These differences in carotenoid contents may be associated with the losses that occur during processing due to crushing the fruits and enzymatic or non-enzymatic oxidative degradation [50].

In pitanga pulp, the highest lycopene content was observed, followed by acerola and the blend. Low lycopene levels were also found by Shukla et al. [51] when evaluating five guava cultivars. They found the presence of lycopene in four, with values ranging from 0.0037 to 0.0332 mg/100 g. Low levels were also found by Leiton-Ramírez et al. [47] in guava slices with a content of 0.0867 mg/100 g. In the red and orange pitangas evaluated by Bagetti et al. [52], values of 0.166 and 0.151 were identified, respectively, lower than the pitanga pulp shown in Table 2.

### 2.3. Colorimetry

In Table 3, the results referring to the color parameters of the three pulps and the elaborated blend are displayed. Significant differences (*p* < 0.05) were observed for all samples in relation to the intensity of yellow. However, for the other parameters, at least two samples did not show heterogeneity.

Regarding color intensity (chroma C*), pitanga and the blend showed some similarity (*p* > 0.05) with the highest values, followed by guava and acerola. The hue angle shows that the four samples are in the same quadrant, with acerola and guava in the range of 0 to 25° (orange red) and pitanga and the blend in the range of 26 to 70° (orange), justified by the fact that the intensity of pitanga’s color is greater than the others, which influences the blend.

Different values of the color parameters were found by Jaeschke et al. [53] when evaluating acerola pulp, obtaining a* = 20.6, b* = 25.0 and L* = 39.9, and in a blend of acerola and pineapple prepared by Silva et al. [54], with results of a* = 35.30, b* = 28.38 and L* = 28.32. This heterogeneity found in relation to the colors depends on several factors, but mainly on the maturation stage, species and variety of the evaluated fruits.

### 2.4. Identification and Quantification of Individual Phenolic Compounds

Table 4 presents the composition of the phenolic compounds of the extracts of the whole pulp of acerola, guava, pitanga and the blend.

In all, nineteen compounds were identified, including six phenolic acids (gallic acid, syringic acid, caftaric acid, chlorogenic acid, caffeic acid and p-coumaric acid), six flavanols (procyanidin B1, procyanidin B2, epigallocatechin gallate, epicatechin, epicatechin gallate and catechin), four flavonols (kaempferol 3-glucoside, rutin, quercetin 3-glucoside and isorhamnetin), one flavone (hesperidin), one anthocyanin (malvidin 3-glucoside) and one stilbene (trans-resveratrol). Of these, eighteen were detected in acerola, nine in guava, twelve in pitanga and fourteen in the blend.

Phenolic acids are the metabolites responsible for the sensory characteristics of fruits, such as color, flavor and astringency, being classified into two subgroups: hydroxybenzoic acid and hydroxycinnamic acid [55]. Of the six identified acids, two (gallic and syringic acid) are hydroxybenzoic and four (caftaric, chlorogenic, p-coumaric and caffeic) are hydroxycinnamic.

Gallic and chlorogenic acids were identified in the four samples, with higher levels in acerola, which had the greatest diversity of acids and the highest levels in relation to the other pulps. Gallic acid had the highest levels in the four analyzed samples (2.20 mg/100 g in acerola, 0.41 mg/100 g in guava, 0.71 mg/100 g in pitanga and 2.17 mg/100 g in the blend). Nascimento et al. [56], studying acerolas at different maturation stages (green, semi-ripe and ripe), also identified gallic (470 to 536 mg/100 g) and caffeic (459 to 682 mg/100 g) acids, with the highest levels appearing in mature acerolas. Gallic acid is a polyphenol found mainly in red fruits and very important for human health due to its high antioxidant, antibacterial and anticancer activity [57]. Chlorogenic acid is one of the most available phenolic compounds in foods [58] and is of great importance, as it is an efficient natural antioxidant and anti-inflammatory, in addition to having broad-spectrum antibacterial activity and certain inhibition of *Escherichia coli*, *Staphylococcus aureus*, yeast, *Aspergillus niger* and *Bacillus subtilis*, as well as good resistance activity against *Staphylococcus aureus* and *Escherichia coli* [59].

The lowest diversity of phenolic acids was identified in guava, with only two, followed by pitanga and the blend, with three. The phenolic acids detected in the blend exceeded the levels found in guava and pitanga. Four phenolic acids were identified by Matos et al. [29] in acerola pulp: 4-hydroxybenzoic acid (1.19 mg/100 g), salicylic (9.50 mg/100 g), trans-cinnamic (1.19 mg/100 g) and p-coumaric (21.37 mg/100 g), which was the only one identified in common with this study. The mixed pulp of jambolan and acerola evaluated by the authors showed two more acids than acerola: 3,4-dihydroxybenzoic acid (23.14 mg/100 g) and vanillic acid (26.61 mg/100 g).

In the seven varieties of acerolas studied by Ferreira et al. [60], three phenolic acids were observed (caffeic –0.03 to 0.06 mg/100 g; trans-caftaric –0.342 to 0.452 mg/100 g; chlorogenic 0.168 to 0.384 mg/100 g), with acid chlorogenic being the only one present in all varieties. Two of these acids (caffeic and chlorogenic) were also identified in the acerola pulp of our study, with values close to those of some varieties.

Flavonoids formed the largest group among those determined, both in quantity and diversity, with acerola also standing out in diversity and quantity, contributing to the blend and also surpassing guava and pitanga pulps. Only two flavonoids (catechin and rutin) were identified by Matos et al. [29] in acerola pulp and in the mixed pulp of jambolan and acerola, with catechin standing out in both samples with 21.96 mg/100 g in acerola and 35.27 mg/100 g in the mixed pulp.

Among the flavanols, epigallocatechin gallate and epicatechin gallate were not detected in guava. The latter was also absent in pitanga. Catechin presented higher levels in all samples, with acerola surpassing the level of other samples.

Four flavonols were found in acerola and three in guava and pitanga, where isorhamnetin was not detected. Kaempferol 3-glucoside was the main flavonol, with the highest levels in pitanga and little expressive value in guava. 

In the flavone group, hesperidin was only detected in acerola and absent in guava and pitanga, while among anthocyanins, only malvidin 3-glucoside was identified and only in pitanga. In stilbenes, trans-resveratrol was found only in acerola, which also contributed to the blend. Ferreira et al. [60], studying the phenolic profile in acerola genotypes, also reported concentrations of trans-resveratrol ranging from 0.234 to 0.385 mg/100 g in addition to four flavanols in common with the acerola pulp in the present work, among which the authors did not identify epigallocatechin gallate and reported the presence of procyanidin A2. The authors found the presence of the same flavonols and flavones plus naringenin.

Much higher values were identified by Nascimento et al. [56] in ripe acerola with 871 mg/100 g of catechin, 704 mg/100 g of epicatechin, 1657 mg/100 g of rutin, 213 mg/100 g of quercetin, 1178 mg/100 g of quercetin and 162 mg/ 100 g of kaempferol, which may be mainly related to the variety of acerola and also growing and processing conditions.

### 2.5. Antioxidant Activity

The results of the antioxidant activity of the pulps and the blend determined using the FRAP, ABTS and DPPH methods are presented in Table 5. From the set of three tests, it is observed that the acerola showed the best performance with the highest antioxidant activities among the pulps, giving the blend good antioxidant activity. In the FRAP assay, the highest values were obtained in all samples in relation to the DPPH and ABTS assays. Guava pulp showed a higher value in the FRAP test than pitanga, while pitanga showed a higher value in the ABTS tests. The variation in antioxidant activity in fruit pulps and in the blend is due to the presence of different phenolic compounds, their concentration and the mechanisms of action of antioxidant assays [43].

Cruz [61] reported a value of 1641.72 µmol/100 mL for DPPH• radical elimination in mature acerola extract. Viana et al. [50] obtained values of 2.72, 5.84 and 4.09 µM Trolox/g for elimination of the ABTS radical in two hybrids and a commercial variety of acerola.

Higher values of antioxidant activity were verified by Matos et al. [44] in the FRAP test for acerola with antioxidant activity of 115.23 mmol/kg, for the mixed pulp of jambolan and acerola with 95 mmol/kg and in the DPPH test with 102.43 mmol/kg and 71.48 mmol/kg, respectively.

### 2.6. Pearson Correlation Analysis between Bioactive Compounds and Antioxidant Activity

To evaluate the interrelationship between bioactive compounds and antioxidant activity in acerola, guava and pitanga pulps and in the blend, Pearson’s correlation analysis between these variables was applied (Figure 1).

It is observed that the ability to scavenge free radicals by DPPH correlated positively, significantly and strongly (r between 0.70–0.89) with the bioactive compounds ascorbic acid (AA), total phenolic compounds (TPC), flavonoids (Flavo) and carotenoids (Carot) and very strong (r between 0.90–1.00) with anthocyanins (Antoci). The correlation of antioxidant activity via FRAP was positive, significant and moderate (r between 0.40–0.69) with the content of ascorbic acid, flavonoids and anthocyanins; strong (r between 0.70–0.89) with TPC and negative, significant and moderate with Licop. For the ABTS assay, the correlation was only significant, positive and strong for carotenoids. Significant, positive and strong correlations of antioxidant activity as indicated via FRAP, ABTS and DPPH with total phenolic compounds and flavonoids content for five peach cultivars were also reported by Zhao et al. [62].

It appears that the DPPH assay correlated more strongly with ascorbic acid, flavonoids, anthocyanins and carotenoids than the other assays, while the FRAP assay correlated better with total phenolic compounds and lycopene. Thus, it is evident that the antioxidant activity of the pulps/blend is mainly due to the phenolic compounds (TPC, flavonoids, anthocyanins and carotenoids) and also to the ascorbic acid content. The antioxidant activity of phenolic compounds is controlled through intermolecular interactions that can be synergistic or antagonistic, depending on the conditions and substances to be evaluated [43].

## 3. Materials and Methods

### 3.1. Raw Material

To carry out this work, acerolas (*Malpighia emarginata*), guavas (*Psidium guajava*) and pitangas (*Eugenia uniflora* L.) were used (Figure 2). They were collected between January and March 2020, in the municipalities of Petrolina (latitude 9°23′39″ S, longitude 40°30′35″ W, altitude 380 m) and Bonito (latitude 8°28′13″ S, longitude 35°43′35″ W, altitude 423 m), both located in the state of Pernambuco in Brazil.

### 3.2. Pulp Extraction and Blend Preparation

The fruits were received and selected through choosing ripe ones with uniform size and free of injuries. They were washed in running water, sanitized via immersion in chlorinated water (50 ppm) for 15 min and immersed in drinking water for rinsing. 

Then, they were pulped in a horizontal mechanical pulper (Laboremus^®^, model DF–200, Campina Grande, Paraíba, Brazil) equipped with a sieve screen with holes of 2.5 mm in diameter. After pulping, the acerola, guava and pitanga pulps were individually packed in low-density polyethylene packages measuring 10 × 25 cm^2^ with a capacity of 200 g and stored in a freezer at a controlled temperature of −18 °C until the moment of the blend preparation.

To obtain the blend, the pulps were mixed with each other in a ratio of 1:1:1 (g/g) and homogenized in a domestic blender (Arno^®^, Power Mix model, Itapevi, São Paulo, Brazil) for 2 min.

### 3.3. Characterization of Pulps and Blend

Whole pulps and the blend were evaluated for physicochemical characteristics, bioactive compounds, profile of phenolic compounds and antioxidant activity.

#### 3.3.1. Physicochemical Characterization

The physicochemical analyses were performed in quadruplicate using the methodologies proposed by the AOAC [63]. The evaluated parameters were water content, through drying in a vacuum oven (QUIMIS^®^, model Q319V, Diadema, São Paulo, Brazil) at 70 °C until constant mass; water activity (a_w_), determined via direct reading at 25 °C in a dew point hygrometer (Aqualab, model 3TE, Decagon Devices^®^), Pullman, Washington, EUA; pH, via direct reading on a digital pHmeter (Tecnal^®^, model TEC-2, Piracicaba, São Paulo, Brazil); total titratable acidity (TTA), via titration with 0.1 mol/L NaOH to pH 8.1; total soluble solids (TSS) in a portable refractometer (Instrutherm^®^, model RT–30 ATC, São Paulo, São Paulo, Brazil); and mineral residue (ash) obtained through calcination in a muffle at 550 °C.

Total sugars (g/100 g) were determined using the methodology of Yemm and Willis [64] and reducing sugars (g/100 g) using the methodology of Miller [65]. Both analyses were performed in a spectrophotometer (Coleman^®^, model 35 D, Santo André, São Paulo, Brazil). Non-reducing sugars were calculated using the difference between total and reducing sugars.

#### 3.3.2. Bioactive Compounds

The ascorbic acid content (mg ascorbic acid/100 g) was determined based on the protocol by Oliveira et al. [66]. The content of total phenolic compounds (TFC) was quantified using the method described by Waterhouse [67]; total carotenoids (g/100 g) were measured according to Lichtenthaler [68]; total flavonoids (mg/100 g) and anthocyanins (mg/100 g) were measured using the methods described by Francis [69]; lycopene was measured according to Nagata and Yamashita [70]. All absorbance readings of the analyses were performed in a spectrophotometer (Coleman^®^, model 35 D, Santo André, São Paulo, Brazil).

#### 3.3.3. Colorimetry

Color parameters were determined in the CIELAB system using a portable colorimeter (MiniScan, Hunter Lab XE Plus, model 4500 L, Hunter Associates Laboratory). The equipment was calibrated with a white plate (X = 80.5; Y = 85.3; Z = 90.0) and equipped with D65 illuminant and observation angle of 10°.

In the CIELAB system, L* represents the brightness, which varies from 0 to 100, with zero corresponding to total black and 100 corresponding to total white; a* is the chromaticity axis that varies from green (−) to red (+) and b* is the chromaticity axis ranging from blue (−) to yellow (+).

In addition to the color coordinates, we calculated the chroma parameters (Equation (1)) representing the purity of the color and the measurement of the hue angle (Equation (2)), which represents the hue of the color.
(1)$$C^{*}=\sqrt{{\left(a^{*}\right)}^{2}+{\left(b^{*}\right)}^{2}}$$
(2)$$h^{*}={tag}^{-1}\left(\frac{b^{*}}{a^{*}}\right)$$

#### 3.3.4. Identification and Quantification of Individual Phenolic Compounds via HPLC

Chromatographic analyses were performed on a high-performance liquid chromatograph (HPLC) (Shimadzu^®^, São Paulo, São Paulo, Brazil), equipped with a Rheodyne 7125i automatic injector and a UV-vis detector. For the identification of individual phenolic compounds, the methodology described by Meireles [71] was used. The columns used were a Shimadzu^®^ model LC-18 column (25 cm × 4.6 mm, particle size 5 µm) (São Paulo, São Paulo, Brazil) and a C-18 ODS Shimadzu^®^ pre-column (São Paulo, São Paulo, Brazil). Samples were eluted with a gradient system consisting of solvent A (2% acetic acid, *v*/*v*) and solvent B (acetonitrile:methanol, 2:1, *v*/*v*), used as mobile phases with a flow rate of 1 mL/min. The column temperature was maintained at 25 °C and the injection volume was 20 µL. The gradient system started from 90% A at 0 min, 88% A at 3 min, 85% A at 6 min, 82% A at 10 min, 80% A at 12 min, 70% A at 15 min, 65% A in 20 min, 60% A in 25 min, 50% A in 30–40 min, 75% A in 42 min and 90% A in 44 min. The total chromatographic run was 50 min. The peaks of phenolic compounds were monitored at 280 nm. Labsolutions software (Shimadzu^®^, São Paulo, São Paulo, Brazil) was used to control the LC-UV and data processing system. Phenolic compounds were identified through comparing retention times with phenolic standards, analyzed under the same conditions, being quantified from calibration curves. The chromatograms were recorded in the Labsolutions Data System software and the results expressed in mg/100 g.

#### 3.3.5. Antioxidant Activity

##### Antioxidant Activity via Ferric Reducing Antioxidant Power (FRAP)

The antioxidant activity according to the ferric reducing power assay was determined using the FRAP method (Ferric Reducing Antioxidant Power) [72] with adaptations. In the absence of light, the FRAP reagent was prepared with 300 mmol/L acetate buffer (pH 3.6), 10 mmol/L 2,4,6-tris (2-pyridyl)-s-triazin (TPTZ) in a solution of 40 mmol/L HCl and 20 mmol/L FeCl3. An aliquot of the sample extract was transferred to a test tube and 0.27 mL of ultrapure water and 2.7 mL of FRAP reagent were added. The mixture was stirred and kept in a bath at 37 ± 1 °C for 30 min. After cooling to room temperature (≈25 °C), samples and standards were read in a spectrophotometer (Shimadzu^®^, model UV-VIS UV-1280, São Paulo, São Paulo, Brazil) at 595 nm. The standard curve was plotted using iron (II) sulfate at concentrations ranging from 500 to 2000 µmol/L. The results were expressed in µmol of ferrous sulfate equivalent/g (µmol Fe^2+^/g).

##### Antioxidant Activity via ABTS•+ Free Radical Scavenging

The ABTS•+ cationic radical was formed through reacting a solution of ABTS (2.2-azinobis (3-ethylbenzthiazoline-6-sulfonic acid)) (7 mM) with potassium persulfate solution (140 mM) incubated at room temperature in the absence of light for 16 h and then diluted in ethanol to an absorbance of 0.70 ± 0.05 nm at 734 nm. The antioxidant activity of the samples was estimated from the mixture of 30 μL of the extract with 3.0 mL of the ABTS•+ radical. The reading was taken after 6 min of the reaction in a spectrophotometer at 734 nm and ethanol was used as a blank. As a reference, Trolox was used to obtain the standard curve, according to the method described by Rufino et al. [73].

##### Antioxidant Activity via DPPH• Free Radical Scavenging

The activity of free radical scavenging based on the DPPH• method was performed according to Rufino et al. [73] with modifications. An aliquot of the extract was added to 3.0 mL of a diluted solution of DPPH• (2.2-diphenyl-1-picryl-hydrazyl) in ethanol (0.0236 mg/mL), stirred and incubated for 30 min in the absence of light, and the absorbance of samples and standards were measured in a spectrophotometer (Shimadzu^®^, model UV-VIS UV-1280) at 517 nm. The standard curve was performed with Trolox (6-hydroxy-2,5,7,8-tetramethylchroman-2-carboxylic acid) (100–2000 µmol/L in ethanol). Results were expressed in µmol Trolox equivalent /g (µmol TE/g).

### 3.4. Statistical Analysis

All data obtained were statistically evaluated with ASSISTAT^®^ software in version 7.7 [74], using a completely randomized design (DIC) and comparing means using Tukey’s test with a significance level of 5%.

Pearson correlation coefficients (r) were calculated between bioactive compounds and antioxidant activity assays using JASP software version 0.17.1 (University of Amsterdam).

## 4. Conclusions

Acerola pulp presented, in relation to guava and pitanga pulps, higher contents of ascorbic acid, total phenolic compounds, total flavonoids, anthocyanins and carotenoids, being surpassed by pitanga pulp in lycopene content. In the acerola pulp, greater diversity and quantity of phenolic compounds were detected, surpassed by guava only in catechin content and by pitanga in kaempferol 3-glucoside and malvidin 3-glucoside contents. Acerola pulp showed the highest antioxidant activity in the evaluation via FRAP, ABTS and DPPH assays. The blend of acerola, guava and pitanga pulps showed positive characteristics conferred by the combination of pulps, with low pH, high levels of total soluble solids and sugars, greater diversity of phenolic compounds than guava and pitanga pulps and antioxidant activity close to that of acerola pulp.

Pearson correlation among antioxidant activity and ascorbic acid content, total phenolic compounds, flavonoids, anthocyanins and carotenoids were positive for the samples. Thus, it is concluded that the tropical red fruit blend is a great food for production and marketing by the food industries, meeting consumers’ need for new sensory experiences and nutrient richness.

## Figures and Tables

**Figure 1 molecules-28-04866-f001:**
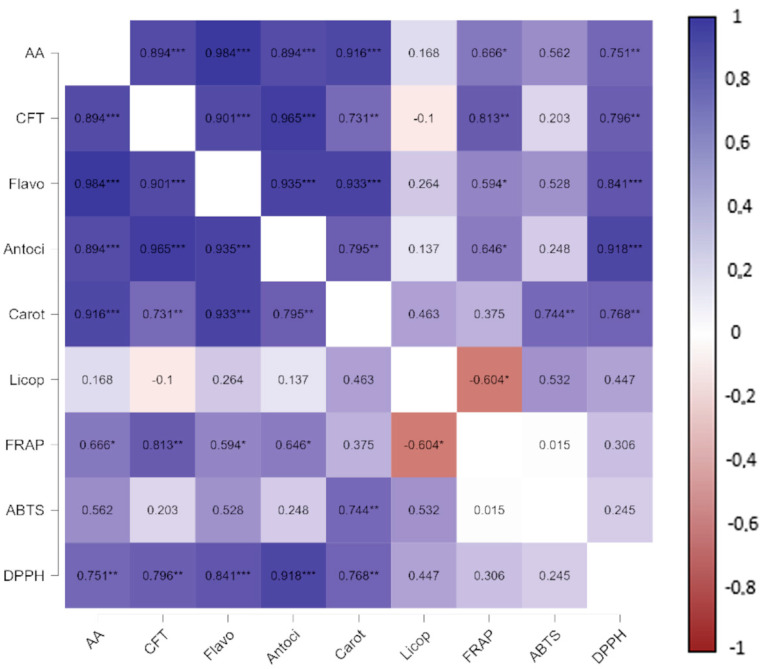
Heat map representing Pearson’s correlation between bioactive compounds and antioxidant activity Significance of Pearson’s correlation: * *p* < 0.05; ** *p* < 0.01; *** *p* < 0.001.

**Figure 2 molecules-28-04866-f002:**
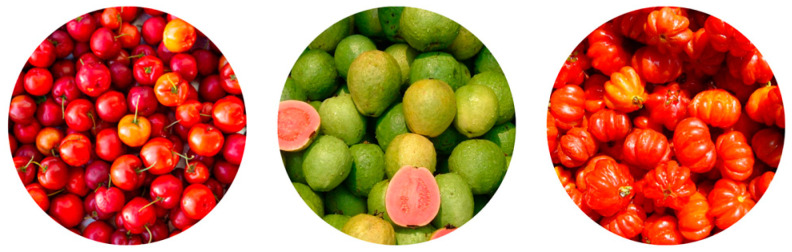
Acerolas (*Malpighia emarginata*), guavas (*Psidium guajava*) and pitangas (*Eugenia uniflora* L.).

**Table 1 molecules-28-04866-t001:** Physicochemical characterization of acerola, guava and pitanga pulp and the blend prepared.

Parameters	Acerola	Guava	Pitanga	Blend
Water content (g/100 g)	92.07 ± 0.01 A	85.50 ± 0.38 C	92.39 ± 0.23 A	89.52 ± 0.28 B
Water activity	0.992 ± 0.00 C	0.996 ± 0.00 AB	0,997 ± 0 A	0.994 ± 0.00 BC
Total titratable acidity (g citric acid/100 g)	2.57 ± 0.05 C	1.29 ± 0.03 D	3.42 ± 0.06 A	2.81 ± 0.05 B
pH	3.52 ± 0.02 B	4.11 ± 0.06 A	3.18 ± 0.08 D	3.39 ± 0.03 C
Ashes (g/100 g)	0.36 ± 0.01 B	0.49 ± 0.02 A	0.30 ± 0.01 C	0.37 ± 0.02 B
Total soluble solids (°Brix)	9.73 ± 0.05 A	10.08 ± 0.05 A	3.60 ± 0.08 B	9.88 ± 0.48 A
Total sugars (g glucose/100 g)	1.25 ± 0.02 C	9.55 ± 0.00 A	0.48 ± 0.00 D	6.09 ± 0.00 B
Reducing sugars (g glucose/100 g)	1.03 ± 0.00 C	9.13 ± 0.01 A	0.44 ± 0.00 D	5.68 ± 0.38 B
Non-reducing sugars (g sucrose/100 g)	0.22 ± 0.02 AB	0.42 ± 0.01 A	0.03 ± 0.00 B	0.41 ± 0.38 A

Means followed by the same letters in the lines do not differ statistically according to Tukey’s test at 5% probability (*p* ≥ 0.05).

**Table 2 molecules-28-04866-t002:** Contents of bioactive compounds in acerola, guava and pitanga pulp and in the blend.

Parameters (mg/100 g)	Acerola	Guava	Pitanga	Blend
Ascorbic acid	3636.43 ± 52.32 A	96,55 ± 1.72 D	289.78 ± 0.87 C	1816.81 ± 30.82 B
Total phenolic compounds	2069.55 ± 0.57 A	309.81 ± 0.10 C	178.63 ± 0.01 D	2027.85 ± 4.32 B
Total flavonoids	12.64 ± 0.50 A	1.91 ± 0.07 D	4.05 ± 0.15 C	8.22 ± 0.21 B
Anthocyanins	2.24 ± 0.09 A	0.48 ± 0.02 B	0.90 ± 0.02 B	2.16 ± 1.11 A
Total carotenoids	7.92 ± 0.39 A	1.69 ± 0.54 D	3.33 ± 1.01 C	4.16 ± 0.95 B
Lycopene	0.19 ± 0.01 B	0.05 ± 0.00 D	0.26 ± 0.02 A	0.10 ± 0.00 C

Means followed by the same letters in the lines do not differ statistically according to Tukey’s test at 5% probability (*p* ≥ 0.05).

**Table 3 molecules-28-04866-t003:** Colorimetric parameters of the acerola, guava and pitanga pulp and the prepared blend.

Parameters	Acerola	Guava	Pitanga	Blend
Brightness (L*)	18.77 ± 0.56 A	12.96 ± 0.32 C	13.35 ± 0.15 C	14.56 ± 0.13 B
Red intensity (+a*)	7.66 ± 0.25 C	10.33 ± 0.69 B	10.87 ± 0.18 B	11.93 ± 0.29 A
Yellow intensity (+b*)	2.44 ± 0.26 D	3.50 ± 0.04 C	8.78 ± 0.21 A	6.63 ± 0.15 B
Chroma (C*)	8,04 ± 0.30 C	10.91 ± 0.65 B	13.97 ± 0.15 A	13.65 ± 0.24 A
Hue angle–h* (o)	17.64 ± 1.33 C	18.76 ± 1.33 C	38.91 ± 0.94 A	29.08 ± 0.94 B

Means followed by the same letters in the lines do not differ statistically according to Tukey’s test at 5% probability (*p* ≥ 0.05).

**Table 4 molecules-28-04866-t004:** Chromatographic profile of phenolic compounds in acerola, guava and pitanga pulp and the blend.

Phenolic Compounds (mg/100 g)	Acerola	Guava	Pitanga	Blend
Phenolic acids				
Gallic acid	2.20 ± 0.60 A	0.41 ± 0.14 B	0.71 ± 0.12 C	2.17 ± 0.55 A
Syringic acid	0.03 ± 0.01	ND	ND	ND
Caftaric acid	0.11 ± 0.01	ND	ND	ND
Chlorogenic Acid	0.15 ± 0.00 A	0.07 ± 0.00 B	0.05 ± 0.00 B	0.13 ± 0.01 A
Caffeic acid	0.06 ± 0.00 A	ND	0.03 ±0.00 B	0.04 ± 0.01 B
p-Coumaric acid	0.03 ± 0.02	ND	ND	ND
∑ Phenolic acids	2.58 A	0.48 D	0.79 C	2.34 B
Flavanols				
Procyanidin B1	0.40± 0.03 A	0.11 ± 0.01 B	0.17 ± 0.03 B	0.25 ± 0.11 AB
Procyanidin B2	0.17 ± 0.04 A	0.09 ± 0.00 A	0.11 ± 0.00 A	0.15 ± 0.02 A
Epigallocatechin gallate	0.32 ± 0.02 A	ND	0.12 ± 0.01 B	0.24 ± 0.05 A
Epicatechin	0.07 ± 0.01 A	0.004 ± 0.01 B	0.03 ± 0.01 B	0.02 ±0.00 B
Epicatechin gallate	0.08 ± 0.00 B	ND	ND	0.10 ± 0.01 A
Catechin	1.14 ± 0.10 A	0.50 ± 0.13 BC	0.17 ± 0.01 C	0.86 ± 0.15 AB
∑ Flavanols	2.18 A	0.70 C	0.60 D	1.62 B
Flavonols				
Kaempferol 3-glicoside	1.39 ± 0.39 A	0.08 ± 0.03 B	2.20 ± 0.22 A	1.76 ± 0.33 A
Rutin	0.18 ± 0.00 A	0.01 ± 0.00 B	0.02 ± 0.00 B	0.07 ± 0.06 AB
Quercetin 3-Glycoside	0.66 ± 0.09 A	0.06 ± 0.00 B	0.18 ± 0.03 B	0.46 ± 0.08 A
Isorhamnetin	0.24 ±0.01 A	ND	ND	0.14 ± 0.03 B
∑ Flavonols	2.47 A	0.15 D	2.14 C	2.43 B
Flavonas				
Hesperidina	0.30 ± 0.05 A	ND	ND	0.47 ± 0.23 A
∑ Flavones	0.30 B	ND	ND	0.47 A
Aanthocyanins				
Malvidin 3-glucoside	ND	ND	0.61 ± 0.00	ND
∑ Anthocyanins	ND	ND	0.61 ± 0.00	ND
∑ Flavonoids	4.95 A	0.854 D	3.35 C	4.52 B
Stilbenes				
Trans-resveratrol	0.10 ± 0.02 A	ND	ND	0.06 ± 0.01 B
∑ Stilbenes	0.10 ± 0.02 A	ND	ND	0.06 ± 0.01 B

Means followed by the same letters in the lines do not differ statistically according Tukey’s test at 5% probability (*p* ≥ 0.05).

**Table 5 molecules-28-04866-t005:** Antioxidant activity according to FRAP, ABTS and DPPH of acerola, guava and pitanga pulp and the prepared blend.

Samples	FRAP (µmol Fe^2+^/g)	ABTS (µmol ET/g)	DPPH (µmol ET/g)
Acerola	21.99 ± 0.02 A	11.14 ± 0.20 A	6.71 ± 0.09 A
Guava	20.15 ± 0.40 B	7.68 ± 0.42 B	5.14 ± 0.04 C
Pitanga	15.63 ± 0.74 C	8.86 ± 0.18 B	6.17 ± 0.01 B
Blend	22.02 ± 0.00 A	6.88 ± 2.01 B	6.76 ± 0.08 A

Means followed by the same letters in the columns do not differ statistically according to Tukey’s test at 5% probability (*p* ≥ 0.05).

## Data Availability

Data can be digitized from the graphs or requested from the corresponding author.

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
