# Peer review of "Physicochemical Aspects, Bioactive Compounds, Phenolic Profile and In Vitro Antioxidant Activity of Tropical Red Fruits and Their Blend"

_molecules, 2023, doi:10.3390/molecules28124866_

Round 1
Reviewer 1 Report
The paper entitled: "Physicochemical aspects, bioactive compounds, phenolic profile and in vitro antioxidant activity of tropical red fruits and their blend" present a "clean" manuscript, with determinations that are common for natural matrices, but may be taken into consideration for a possible publication, because in my opinion such studies are important from consumer point of view, more that scientific soundness.
However, I have some minor suggestions for the authors:
- Change the Chapter 2 name from "Results" in "Results and Discussions"
- Use in all Tables only "." when writing the values, not mix "." and ","
- Use the same number of decimals when writing values or standard deviation numbers
- Instead of writing as footer the measurement units in Table 1, better write them near every parameter, and let as footer only statistics and eventually : "mean values represent ... replicates measurement"
English language is ok, please check again some minor errors
Author Response
Dear reviewer
We greatly appreciate your contributions to our manuscript to acquire more quality. I would like to inform you that all revisions have been carried out.
We are available for any questions or others suggestions.
Yours sincerely
Reviewer 2 Report
The manuscript is well written. I would suggest the authors change the order of chapters. The material and method should come before results. Further, I would recommend adding an overall evaluation of the fruits - for example, using multivariate statistical methods (there are many results, so it would be interesting for the readers).
Author Response
Dear reviewer
We greatly appreciate your contributions to our manuscript to acquire more quality. I would like to inform you that the order of the chapters was written following the journal's model, so the material and methods part comes after the results.
Regarding the multivariate statistical methods, Pearson's Correlation was performed, which is already included in the article, but we appreciate the recommendation and will take it into account for future articles to add more analyzes of this type.
We are available for any questions or other suggestions.
Yours sincerely
Reviewer 3 Report
The authors present an unusual study of fruit composition and study of antioxidant properties of fruit extracts. The manuscript may be of interest to a wide range of readers and may be accepted for publication after correction of remarks.
The main issues concern the design of the results.
From the reviewer's point of view, a photograph of the fruit under study should have been added to make it more visible to the reader.
The data tables are extremely poorly designed. so, there are many typos in the data with numbers, it is not quite clear how to interpret the numerical data. All letters in the data should be in upper case.
In some cases the error of 0.00 is confusing. How many independent samples were examined in each case?
The long sentence on page 5, lines 192-201 is unfortunate. it should be redone.
The authors need to be clear in their conclusions about what they found in their work.
The list of references is incorrect and should be corrected.
Author Response
Dear reviewer
We greatly appreciate your contributions to our manuscript to acquire more quality. I would like to inform you that all revisions have been carried out.
Five independent samples were used, with four replicates in each case. Offsets are only to two decimal places, so only 0.00 appears.
We are available for any questions or others suggestions.
Yours sincerely